# Mitochondrial Protection by PARP Inhibition

**DOI:** 10.3390/ijms21082767

**Published:** 2020-04-16

**Authors:** Ferenc Gallyas Jr., Balazs Sumegi

**Affiliations:** 1Department of Biochemistry and Medical Chemistry, University of Pecs Medical School, 7624 Pecs, Hungary; balazs.sumegi@aok.pte.hu; 2Szentagothai Research Centre, University of Pecs, 7624 Pecs, Hungary; 3HAS-UP Nuclear-Mitochondrial Interactions Research Group, 1245 Budapest, Hungary

**Keywords:** cell death, apoptosis, mPT, AIF, ROS, Akt, MAPK

## Abstract

Inhibitors of the nuclear DNA damage sensor and signalling enzyme poly(ADP-ribose) polymerase (PARP) have recently been introduced in the therapy of cancers deficient in double-strand DNA break repair systems, and ongoing clinical trials aim to extend their use from other forms of cancer non-responsive to conventional treatments. Additionally, PARP inhibitors were suggested to be repurposed for oxidative stress-associated non-oncological diseases resulting in a devastating outcome, or requiring acute treatment. Their well-documented mitochondria- and cytoprotective effects form the basis of PARP inhibitors’ therapeutic use for non-oncological diseases, yet can limit their efficacy in the treatment of cancers. A better understanding of the processes involved in their protective effects may improve the PARP inhibitors’ therapeutic potential in the non-oncological indications. To this end, we endeavoured to summarise the basic features regarding mitochondrial structure and function, review the major PARP activation-induced cellular processes leading to mitochondrial damage, and discuss the role of PARP inhibition-mediated mitochondrial protection in several oxidative stress-associated diseases.

## 1. Introduction

Inhibitors of the nuclear DNA damage sensor and signalling enzyme poly(ADP-ribose) (PAR) polymerase (PARP) have recently gained considerable interest in tumour therapy [1]. Following the success of clinical trials, the United States Food and Drug Administration (FDA) approved four PARP inhibitors in the therapy of breast cancer gene (BRCA)1/2 mutated malignancies not responding to conventional treatments [2], and further trials are currently ongoing to extend their use to cancers carrying other forms of genomic instability [3]. Additionally, based on their cytoprotective properties demonstrated in many models, PARP inhibitors were suggested to be repurposed for non-oncological diseases such as acute pancreatitis, stroke, lung injury, traumatic brain injury, septic shock, and various degenerative diseases [4].

PARP1, the family’s principal member, is present in a million-copy number in the nucleus of mammalian cells [5]. When PARP1 is incapacitated, PARP2 takes over its role and vice versa. Together, they are assumed to perform more than 90% of DNA strand break-induced PARylation [5]. PARP1- and PARP2-deficient mice are viable and their phenotype does not considerably differ from that of the wild type. However, the PARP1/PARP2 double knock out mutation results in embryonic lethality [5]. PARP1 regulates various cellular processes either by its enzymatic activity or simply by its physical presence via protein–protein interactions [6]. Several of these processes lead to the destabilization of mitochondrial membrane systems and impairment of mitochondrial function that eventually can result in cell death [7]. Therefore, inhibitors of the enzyme protect mitochondrial integrity and function, yet may contribute to PARP inhibitor resistance, which limits cancer therapy [8]. Preclinical evidence for the PARP inhibitors’ potential in non-oncological human diseases were reviewed elsewhere consistently [4]. Therefore, in this review, we focused on the processes involved in the PARP inhibitors’ protective effects. We summarised the basic features of mitochondrial structure and function, reviewed the major PARP activation-induced cellular processes leading to mitochondrial damage, and presented mechanistic aspects of PARP activation-induced mitochondrial damages in several oxidative stress-associated diseases.

## 2. Mitochondrial Structure and Function

### 2.1. Mitochondria, Mitochondrial DNA, and Oxidative Phosphorylation

Individual mitochondria have a spherocylindrical shape of approximately 1–2 µm in length and 0.1–0.5 µm in diameter [9]. They are characterised by two membrane systems; the outer membrane encloses the intermembrane space, and the highly folded inner membrane envelops the matrix. The outer membrane’s protein to phospholipid ratio is nearly 1:1 by weight, and it contains a substantial number of pore-forming voltage-dependent anion channels (VDACs), which, in their open state, allow the passage of small molecules across the membrane. The inner membrane contains roughly three times as much protein by weight when compared with the outer one, and is rich in cardiolipin, which makes this membrane essentially impermeable to all substances with the exception of gases [10]. Channels and specific transporters are responsible for trafficking substrates and products of the mitochondrial metabolic processes [11]. The matrix contains several copies of the mitochondrial DNA as well as a substantial number of different enzymes, including those of pyruvate and fatty acid oxidation, and also those of the Krebs cycle [9]. The human mitochondrial genome of 16,569 DNA base pairs contains at least 37 genes, which encodes 13 proteins of the oxidative phosphorylation machinery, 22 tRNAs, and 2 rRNAs [12]. Mitochondrial oxidative phosphorylation produces most of the ATP necessary for different functions of the various tissues. The process’ machinery, the enzyme complexes of the respiratory chain, are located in the inner mitochondrial membrane, where they receive the reduced dinucleotides NADH and FADH_2_ resulting from the breakdown reactions of the fuel molecules including glucose, amino acids, and lipids. Electrons of the reduced dinucleotides via stepwise redox reactions are transferred along the respiratory complexes I to IV, and, eventually, reduce molecular oxygen. According to the reigning chemiosmotic hypothesis, simultaneously to the electron transport, a proton gradient is built up between the matrix and the intermembrane space, which represents the driving force for ATP synthesis from ADP and phosphate at complex V [13]. Regardless of the exact mechanism, oxidative phosphorylation yields close to eight times as much ATP per glucose molecule when compared to substrate-level phosphorylation [13,14,15].

### 2.2. Mitochondrial Membrane Potential

Mitochondrial membrane potential (ΔΨm) results from the electrical (charge difference) potential of the ions’ gradient across the mitochondrial inner membrane. In addition to contributing to the proton motive force regarding ATP synthesis, ΔΨm is considered to have a substantial role in mitochondrial translocation of proteins encoded by the nuclear DNA [16], transport of metal ions, such as K^+^, Ca^2+^, and Mg^2+^ [17,18,19], regulation of reactive oxygen species (ROS) production by the mitochondrial electron transport chain [20], regulation of mitochondrial network dynamics [21], and most importantly, regulation of cell death [22,23,24]. These mitochondrial phenomena are intricately interconnected. Mitochondrial biogenesis as well as replacement of proteins, which are continuously damaged by mitochondrially produced ROS, requires the concerted operation of the nuclear and mitochondrial gene expression [25]. Additionally, nearly 50 to 70% of the nuclearly encoded mitochondrial proteins possess positively charged N-terminal mitochondria-targeted sequences, and transport of these proteins via translocases of the inner/outer membrane (TIM/TOM complex) is regulated by ΔΨm [16,26]. K^+^ via the K^+^/H^+^ antiporter and the ATP-sensitive K^+^ channel prevents excess matrix swelling and contraction, respectively, thereby maintaining activity of the electron transport chain, and regulating mitochondrial ROS production [27,28]. Intracellular Ca^2+^ enters the matrix along its electrochemical gradient via the Ca^2+^ uniporter route when its concentration exceeds a submicromolar threshold level [29,30], and the accumulated mitochondrial Ca^2+^ is released via a Na^+^/Ca^2+^ exchanger when the intracellular Ca^2+^ signal declines [31].

### 2.3. Mitochondrial Ca^2+^ and ROS

Elevated mitochondrial Ca^2+^ causes allosteric stimulation of pyruvate-, isocitrate-, and α-ketoglutarate dehydrogenase complexes [32], as well as the activation of complex V [33], glycerol-3-phosphate dehydrogenase [34], and adenine nucleotide translocase [35]. Eventually, this coordinated activation of the mitochondrial energy-producing metabolic pathways results in increased ATP synthesis [32]. However, the increased metabolic rate is accompanied by higher respiratory chain electron leakage that may result in more mitochondrial ROS generation by complexes I and III [36]. In addition, Ca^2+^ by stimulating nitric oxide synthase (NOS) [37], can induce the formation of NO**˙**, which inhibits complex IV [38] that enhances ROS generation at complex III [39]. Comprehensively, increased mitochondrial Ca^2+^ seemingly promotes ROS production in the presence of respiratory chain inhibition while attenuates it under normal conditions [40]. The predominant mitochondrial ROS produced by the respiratory chain is O_2_^‾^**˙** that a spontaneous chemical reaction or the enzymatic action of superoxide dismutase can convert to H_2_O_2_, which can be transformed to OH**˙** by metal ions in the Fenton reaction [41]. Mitochondrial ROS generation is strongly dependent on ΔΨm since mild uncoupling was demonstrated to reduce mitochondrial ROS production in various systems [42,43,44,45].

### 2.4. Mitochondrial Permeability Transition

Mitochondrial Ca^2+^ overload, ROS, and elevated inorganic phosphate initiates mitochondrial permeability transition (mPT). During the process, the permeability transition pore (PTP), a high conductance cyclosporine A sensitive channel of the inner mitochondrial membrane opens and allows rapid unselective passage of water and solutes up to 1.5 kDa of size across the inner membrane [22,46]. The assumed physiological opening of the PTP allows the rapid exchange of solutes, such as Ca^2+^ and ROS, between the matrix and the cytosol [47]. This mechanism was proposed to supplement the mitochondrial Na^+^/Ca^2+^ exchanger in removing excess Ca^2+^ from the matrix once the cytosolic Ca^2+^ decreases below its submicromolar threshold level after a transient elevation [48]. However, the long-term opening of the PTP results in dissipation of ΔΨm, the reversal of ATP synthase’s operation, in an attempt to maintain ΔΨm [49], excess ROS production, release of Ca^2+^ from the matrix and of proapoptotic proteins from the intermembrane space leading to apoptotic cell death, and influx of water into the matrix, which causes mitochondrial swelling and rupture of the inner membrane resulting in necrotic cell death [50]. Operation of mPT in this manner was implicated in the pathomechanism of various diseases including myocardial infarct [51], stroke [52], neurodegenerative diseases [53], and muscular dystrophy [54].

### 2.5. Mitochondrial Fission and Fusion

Mitochondria of a cell continually oscillate between the forms of individual entities and the fused network of multiple organelles by the processes of fusion and fission [55]. These processes have roles in such fundamental functions as mitochondrial biogenesis [56], cellular propagation of mitochondria, and mitochondrial quality control [57,58]. Fragmentation of the mitochondrial network occurs in response to impairment of oxidative phosphorylation once the mitochondrial inner membrane potential is abnormal [59,60]. However, fragmentation of mitochondria does not necessarily evoke apoptosis [55]. The mitochondrial outer membrane fission protein (FIS)-1, the limiting factor in the fission process, recruits dynamin-like protein (DRP)-1 to punctuate structures on the surface of the mitochondria. DRP-1 homo-oligomers perform the fission by forming a ring around the mitochondrial tubule, and generate mechanical force via conformational changes leading to membrane constriction, similar to dynamin [61]. In contrast to fission, fusion needs separate dynamin-like large GTPases for the sequential fusion of the outer and the inner mitochondrial membranes [62]. Fusion of the outer membrane is mediated by the resident large GTPases mitofusin (Mfn)1 and 2 [63], both of which were found to be essential regarding embryonic development [64]. Mfn1 is expressed ubiquitously, and it has a higher GTPase activity when compared to Mfn2 [65]. Expression of Mfn2 is tissue-specific [66], and it was reported to have a role in tethering mitochondria and endoplasmic reticulum [67]. Fusion of the inner membrane is mediated by another dynamin-like large GTPase, optic atrophy (OPA)1 [62]. The enzyme exists in two forms, an inner membrane-anchored long form required for fusion, and a soluble short form associated with fission rather than fusion of the inner membrane. Although differential splicing can account for the existence of the two forms [68], activation of the metalloendopeptidase OMA1 under stress conditions can lead to complete conversion of the long to the short form of OPA1, which prevents mitochondrial fusion until the *de novo* synthesis of new OPA1 [69].

## 3. Interplay of PARP with Akt-Mediated Mitochondrial Protection

### 3.1. Akt’s Effects on Outer Mitochondrial Membrane Permeabilisation-Associated Processes

The phosphatidylinositol-3 kinase (PI3K)-protein kinase B/Akt pathway mediates proliferation-inducing and cytoprotective effects of cAMP, hypoxia and cytokines, as well as various growth factors [70]. The activation of the pathway can protect the mitochondria via various mechanisms including preservation of the outer mitochondrial membrane’s integrity [71]. Pro-apoptotic and anti-apoptotic members of the B-cell lymphoma (Bcl)-2 protein family have opposite effects on the outer membrane. Heterodimerisation of pro-apoptotic members, such as Bcl-2-associated X (Bax) and Bcl-2 homologous antagonist/killer (Bak), especially in the presence of Bcl-2 homology domain (BH)3-only proteins, such as Bcl-2-associated agonist of cell death (Bad), Bcl-2-like protein 11 (Bim), BH3 interacting-domain death agonist (Bid) and p53 upregulated modulator of apoptosis (PUMA) permeabilises the outer membrane via pore formation, which allows cytochrome C release from the mitochondrial intermembrane space leading to caspase-9 activation that eventually results in apoptotic cell death [71,72]. Anti-apoptotic members, such as Bcl-2, Bcl-xL, and myeloid leukemia cell differentiation protein (Mcl)-1, antagonise the pro-apoptotic family members’ effect thereby preserving the outer membrane’s integrity and promoting cell survival [73]. Akt phosphorylates Bad, which forestalls the heterodimer formation between Bad and other pro-apoptotic Bcl-2 family members. Rather, phosphorylated Bad, by forming a complex with the cytoplasmic scaffolding protein 14-3-3, is eliminated from the balance between the pro-and anti-apoptotic Bcl-2 family members, resulting in the prevention of cytochrome C release [74]. Additionally, Akt directly phosphorylates, and thereby inactivates caspase-9 at its Ser^196^, which contributes to Akt’s effect of blocking the intrinsic apoptotic pathway [75].

### 3.2. Akt’s Effects on Glycogen Synthase Kinase-3β-Mediated Processes

Due to its extensive participation in signalling pathways (in addition to its metabolic role of regulating glycogen synthesis), glycogen synthase kinase (GSK)-3β is one of Akt’s most prominent downstream targets in mediating mitochondrial protection [76]. In hypoxia, GSK-3β is activated by phosphorylation of its Tyr^216^ [77], and contributes to the hypoxia or ischemia-induced tissue injury. It represses expression, nuclear translocation, and binding to antioxidant response element (ARE) DNA sequence, of nuclear factor erythroid 2-related factor 2 (Nrf2) [78]. The diminished binding of the transcription factor results in a reduced expression of Nrf2/ARE-regulated genes encoding proteins of the antioxidant defence system, such as superoxide dismutase, peroxidase, catalase, and enzymes of glutathione synthesis and reactivation [78]. The resulting oxidative stress damages the mitochondria, which significantly contributes to hypoxia or ischemia-induced mitochondrial impairments and tissue injury [76]. Furthermore, the activation of GSK-3β diminishes nuclear translocation of the transcription factor cAMP response element-binding protein (CREB), thereby diminishing CREB’s binding to the co-activator CREB-binding protein (CBP) [79]. This process causes altered interaction with the pro-inflammatory transcription factor nuclear factor (NF)κB leading to increased inflammatory response [80]. The oxidative stress accompanying the inflammatory response contributes to mitochondrial damages [76]. Akt counteracts these harmful effects of GSK-3β by phosphorylating its Ser^9^, thereby inhibiting the enzyme [81]. In addition to preventing GSK-3β’s diminishing effect on CREB activation [79], Akt directly phosphorylates CREB’s Ser^133^, which promotes CREB’s binding to CBP and enhances expression of CREB-regulated genes critical for mitochondrial protection and survival [82].

### 3.3. Akt’s Effects on Mechanistic Target of Rapamycin and Forkhead Transcription Factor-Mediated Processes

The other major node of the PI3K-Akt pathway is mechanistic (previously mammalian) target of rapamycin (mTOR), a downstream target of Akt, mTOR complex (mTORC)1, and an upstream activator of Akt, mTORC2. Akt activates mTOR by direct phosphorylation, which induces transcription factors associated with growth and cell survival, as well as factors regulating translation initiation, hypoxia, and angiogenesis [83]. Furthermore, the mTOR pathway activates peroxisome proliferator–activated receptor γ coactivator-1α (PGC-1α), the major transcription factor of mitochondrial biogenesis, thereby modulating the mitochondrial copy number and mitochondrial function [84,85]. In addition to phosphorylating its cytoplasmic targets, activated Akt translocates to the nucleus, and regulates various transcription factors by phosphorylating them [86]. Forkhead family transcription factors induce the expression of genes, which encode various growth factors, proteins involved in the stress response, and synthetic enzymes of carbohydrate and lipid metabolism [87]. Additionally, Bcl-2 family members can be transactivated by forkhead transcriptional factors. Two functional forkhead response elements were reported to be present within the sequence of Bim promoter [88,89]. When they are phosphorylated by Akt, forkhead transcriptional factors do not translocate to the nucleus, rather, they form a complex with the cytoplasmic 14-3-3 protein and are subjected to proteosomal degradation [87].

### 3.4. Mechanism for Akt-Mediated Protective Effect of PARP Inhibition

Negative regulation of the PI3K-Akt pathway is mediated by the phosphatase and tensin homologue deleted on chromosome 10 (PTEN) [90], an oxidation-sensitive SH enzyme [91]. Due to PTEN’s rapid oxidative inactivation, most oxidative stress situations activate Akt for a period of time, although this activation is usually insufficient to protect the cells against the oxidative stress-induced mitochondrial and cellular damages [92,93]. Via generating DNA strand breaks, oxidative stress activates the PARP, which counteracts the Akt activation and potentiates the ROS-induced impairment of cellular and mitochondrial macromolecules. Therefore, PARP inhibitors possess a potential to protect the macromolecules against oxidative damages [94]. As it was demonstrated in a number of in vitro, ex vivo and in vivo experiments, nuclear PARP inhibition achieved either by pharmacological means or by gene silencing, activates Akt [92,95,96]. This activation could become so substantial that it significantly contributes to the mitochondria- and cytoprotective effects of PARP inhibitors in experimental or pathological oxidative stress situations [97,98,99,100,101]. Recently, we presented a mechanism (Figure 1) for PARP inhibition-induced Akt activation [102]. It proposes that the oxidative stress-generated DNA strand breaks activate PARP1. The enzyme PARylates itself, which forms a scaffold, and recruits ataxia telangiectasia mutated kinase (ATM) and nuclear inhibitor of NF-κB kinase-γ (NEMO) [103]. The attachment of ATM to the PAR scaffold results in its activation and phosphorylation of itself and of NEMO [104]. However, when a pharmacological inhibitor or genetic manipulation precludes excess PARP activation, the unPARylated ATM complexes with NEMO and translocates to the cytoplasm, where the complex binds to the outer mitochondrial membrane, at least partially. There, the ATM–NEMO complex forms a signallosome with Akt and mTOR that activates Akt, which, in turn, can exert its mitochondria- and cytoprotective effects via its downstream targets (Figure 1) [102]. This model contains several elements already reported by others. ATM was reported to mediate Akt activation in cancer cell lines, in insulin-treated myocytes, and under cellular stress conditions [105,106,107]. The pathomechanism of diabetes and neurodegenerative diseases was found to involve impaired ATM-related Akt activation [107]. Furthermore, as it was formerly published by independent groups, PARP1 can PARylate ATM directly in different oxidative stress situations [108,109]. Finally, the process of ATM–NEMO complex formation followed by its nuclear-to-cytosolic translocation in response to PARP activity has already been published as a fundamental part of the mechanism of genotoxicity-induced NFκB-regulated apoptosis [103,110]. To complement the elements others formerly described, we demonstrated how silencing any component of the proposed ATM–NEMO–Akt–mTOR signallosome has abolished the PARP inhibitor’s protective effect, and also, that the signallosome is localised to the mitochondria, at least partially (Figure 1) [102]. We also demonstrated that, although PARP1 inhibition causes synthetic lethality in ATM-deficient cells [111], transdominant transfection of these cells with continuously active Akt rescued them, emphasising the significance of Akt activation in the protective effect of PARP inhibition [102]. This protective effect is beneficial regarding the therapy of non-malignant diseases, yet impairs the therapeutic efficacy of the PARP inhibitors in cancer [112]. The latter concept is confirmed by independent observations, in which Akt inhibitors increase toxicity of PARP inhibitors in cancer cells [113,114,115].

## 4. Interplay of PARP with MAPK-Mediated Mitochondrial Damage

### 4.1. Mitogen-Activated Protein Kinases and Their Activation in Oxidative Stress

Mitogen-activated protein kinase (MAPK) cascades transmit extracellular signals to intracellular targets thereby regulating a number of various cellular processes including proliferation, differentiation, apoptosis and stress responses [116]. The cascades consist of 2-3 layers of upstream activating kinases, the six MAPK groups (extracellular signal-regulated kinase (ERK)1/2, the c-Jun N-terminal kinases (JNK), the p38 MAPKs, ERK3/4, ERK5 and ERK7/8), and their downstream targets. In most systems, the apoptosis signal-regulating kinase (ASK)1/ MAPK kinase kinase (MEKK)1/2—dual specificity MAPK kinases 4 and 7 (MKK4/7)—JNK and ASK1/dual leucine zipper-bearing kinase (DLK)/MEKK3/4—MKK3/6—p38 MAPK pathways are related to stress responses and apoptosis, while the rapidly accelerated fibrosarcoma protein (Raf)—MAPK kinase (MEK)—ERK signalling pathways are associated with proliferation and differentiation [117]. Depending on the tissue type, a number of different signals can activate the MAPK cascades. However, they are readily activated by ROS in most systems [118]. Various mechanisms induce this activation. Phosphatases, responsible for inactivation of the upstream activating kinases, are oxidation-sensitive and rapidly break down in oxidative stress. The ROS can act directly on growth receptors in the absence of their ligands, thereby inducing Ras activation. Furthermore, ROS is capable of activating Ras in a receptor-independent manner [119], and stimulating ERK1/2 activation even in Ras negative cells via non-receptor protein tyrosine kinase c-Src [120]. Oxidative stress and reactive nitrogen species (RNS) activate JNK and p38 pathways through ASK1 and MEKK1. The ROS oxidises and/or the RNS nitrosylates reduced thioredoxin, which inhibits ASK1 by binding to its N-terminal. The oxidised/nitrosylated thioredoxin disassociates, which enables ASK1 oligomerisation and autophosphorylation resulting in activation [121]. Additionally, pro-inflammatory cytokines, such as tumour necrosis factor-α can also activate JNK and p38 MAPK pathways [122].

### 4.2. MAPK-Mediated Mitochondrial Processes

When activated, ERK1/2 supports ATP synthesis, preserves ΔΨm, forestalls cytochrome C release, and inactivates Bad by activating p90 ribosomal S6 kinase, which phosphorylates Ser^155^ of Bad, thereby inactivating it [123,124]. In contrast to these mitochondria protecting effects, ERK was reported to mediate oxidative damage, such as vacuolation, mitochondrial translocation of Bax, and outer membrane permeabilisation to neuronal mitochondria, seen in neurodegenerative diseases [125]. Also, activated ERK1/2 is targeted to the mitochondria, where it associates with the outer mitochondrial membrane, and seemingly enters the intermembrane space to be in position to exert its benign or malign effects [126]. In contrast to ERK1/2, JNK and p38 MAPK activation are implicated nearly exclusively to be involved in mediating mitochondrial damage in various oxidative stress situations. They facilitate mitochondrial Bax translocation, transcriptional regulation of TR-3, an apoptosis-initiating steroid receptor-like protein, cytochrome C and the second mitochondria-derived activator of caspase (Smac) release, and activating phosphorylation of Bad [127]. JNK also phosphorylates the anti-apoptotic Bcl-2 family member Bcl-xL on its Thr^47^ and Ser^62^, which decreases Bcl-xL’s binding to Bax thereby impairing anti-apoptotic function of Bcl-xL [128]. JNK but not p38 MAPK signalling is seemingly associated with mitochondrial ROS generation. The elevated mitochondrial ROS together with Ca^2+^ overload and ATP depletion induces mPT, which may aggravate JNK activation-mediated mitochondrial dysfunction [129]. Similarly to ERK1/2, mitochondrial targeting of JNK may likely be essential for exerting its effects [130]. Recently, a mitochondrial SH3-domain-binding protein 5 (SAB) was reported to operate as the only mitochondrial docking site for JNK [127]. SAB is localised to the outer membrane and binds JNK with its N-terminal SH3 domain, which is in the mitochondrial intermembrane space. SAB depletion prevents mitochondrial translocation of JNK completely. However, it does not interfere with p38 MAPK’s association with the mitochondria [131]. In addition to its apoptosis-inducing effects, mitochondrial JNK regulates mitochondrial bioenergetics, decreases respiration rates and ATP production, and by phosphorylating pyruvate dehydrogenase, shifting the metabolism from aerobic toward anaerobic [129].

### 4.3. Mechanism for MAPK Phosphatase-1-Mediated Protective Effect of PARP Inhibition

As it was previously reported, ROS or alkylating agent-induced PARP activation leads to the collapse of ΔΨm, induction of mPT, and, inevitably, cell death [132,133,134,135]. Notably, these detrimental effects are often mediated by JNK and p38 MAPK activation [134,135,136]. Furthermore, activation of MAPK phosphatase (MKP)-1, which dephosphorylates thereby inactivates at least all classical MAPK groups such as ERK1/2, JNKs and p38 MAPKs, was reported to ameliorate the damages due to oxidative stress [137,138]. Accordingly, PARP inhibition reduces the detrimental effects of oxidative stress, and upregulates MKP-1 expression as we previously discovered [138]. Since MKP-1 expression is regulated by heat shock and cAMP-mediated mechanisms [139,140], after systematically investigating how silencing all pertinent heat shock factors and CREB transcription factors affects PARP inhibition-induced MKP-1 expression, we determined that activating transcription factor (ATF)4/CREB2 is responsible for mediating the MKP-1-upregulating effect of PARP inhibition [141]. Accordingly, we proposed a mechanism (Figure 2) consistent with all experimental data regarding the mitochondria- and cytoprotective effects of PARP inhibition in oxidative stress [141]. Accordingly, ROS-induced DNA strand breaks activate PARP1, which PARylates ATF4 transcription factor. PARylation reduces ATF4′s affinity toward its responsive elements and is replaced to self-PARylated PARP-1. The resulting diminished MKP-1 expression fails to compensate for the MAPK-activating effect of oxidative stress that leads to MAPK activation, which causes mitochondrial damage and ultimately, cell death. Inhibition of PARP either by a pharmacological agent or genetic manipulation, turns the entire process around resulting in mitochondrial protection and cell survival (Figure 2) [141]. All steps of the model were extensively validated. ROS-induced PARP activation was shown to cause massive PARylation and reduced binding of ATF4, while increased binding of PARylated PARP to ATF4′s target DNA sequence. The PARP inhibitor PJ-34 turned all these effects around, while neither ATF4 nor PARP1 demonstrated affinity toward a mutated CRE sequence under any of the experimental conditions, indicating specificity of the DNA binding. Furthermore, PJ-34 strengthened ATF4 binding to MKP-1′s initiation site. Accordingly, inhibition, or silencing of PARP-1 upregulated MKP-1 expression both in the nucleus and the cytoplasm, inactivated JNK and p38 MAPK, preserved ΔΨm, and reduced the cell death. Most importantly, ATF4 and/or MKP-1 silencing forestalled these effects of the PARP inhibitor [141]. Similarly to the case of Akt activation, JNK and p38 MAPK inhibition by the PARP inhibitor, and the resulting mitochondria- and cytoprotection are beneficial in the therapy of non-malignant diseases, yet impairs the PARP inhibitors’ therapeutic efficacy in cancer [112].

## 5. Interplay of PARPs with Mitochondrial Homeostasis

### 5.1. Role of NAD^+^ Metabolism in Mediating PARP Activation-Induced Damages

Oxidative stress situations causing excessive DNA damage activate nuclear PARP, which leads to the collapse of ΔΨm, reduced activity of respiratory complex I, diminished mitochondrial oxidation and ATP production, enhanced O_2_^‾^**˙** production, and destruction of mitochondrial architecture [142,143]. The elevated ROS, Ca^2+^ overload, and ATP shortage open the mitochondrial PTP, which results in the release of the mitochondrial content, swelling due to water influx into the matrix, and even disruption of the mitochondria [144]. Short mPT events can result in cytochrome C and Smac release from the intermembrane space. In the cytosol, cytochrome-C associates with apoptotic peptidase-activating factor (Apaf)1 and procaspase-9 that activates caspase-9 by proteolytic cleavage, which induces effector caspases such as caspase-3 leading to caspase-mediated apoptosis [145]. However, the short-term opening of the PTP more often causes translocation of apoptosis-inducing factor (AIF) and endonuclease G from the mitochondria to the nucleus leading to parthanatos, a special type of programmed necrosis [146]. In the so-called ‘suicide hypothesis’ presented more than 35 years ago, cell death resulted from PARP overactivation was attributed to energy failure. According to the hypothesis, rapid consumption of the substrate NAD^+^ by the copious PARP enzyme depletes cellular NAD^+^ pools, then ATP is consumed in a failed attempt to re-synthesise NAD^+^ [147]. Indeed, the mitochondrial NAD^+^ pool can represent 40% to 70% of the cell’s total NAD^+^ content depending on the cell type. However, the mitochondrial inner membrane is impermeable for NAD^+^, therefore the mitochondrial pool is inaccessible for the nuclear PARP1, unless mPT or necrotic disruption of the mitochondria occur [148]. Factually, in ischemia–reperfusion of isolated heart tissue where massive NAD^+^ loss was observed, cyclosporin A, an mPT inhibitor was reported to prevent the NAD^+^ loss as well as the cardiomyocyte damages indicating the pivotal role of PTP opening in the detrimental consequences of PARP activation in this system [142,149]. On the other hand, in neuronal systems, nuclear PARP1 activation may induce mitochondrial dysfunction without the depletion of the mitochondrial NAD^+^ pool. The glycolytic enzyme glyceraldehyde-3-phosphate dehydrogenase requires NAD^+^ for its operation. Therefore, reduction of cytosolic NAD^+^ pool by excess PARP1 activation can forestall the entry of glucose-derived acetyl-coenzyme A to the matrix [148]. It implies an insufficient supply of the key substrate for the citrate cycle, resulting in energy starvation and mitochondrial depolarisation, since the brain relies on glucose as a fuel in support of mitochondrial ATP production [150]. During the process of parthanatos, following PAR polymer production by excess activation of PARP1, PAR fragments produced by PAR glycohydrolase and ADP ribosyl-acceptor hydrolase can act directly on the mitochondrial membranes to induce AIF release independent of NAD^+^ depletion [151]. Further degradation of the PAR fragments by ADP-ribose pyrophosphatase produces AMP, which can inhibit adenine nucleotide translocase, further impeding mitochondrial energy production [143]. High AMP may activate AMP-activated protein kinase (AMPK), which can phosphorylate, and thereby activate PARP1. Since PARP1 also activates AMPK, together, they form a feed-forward loop [152].

### 5.2. Metabolic Changes in Ischemia

In cultured neurons and astrocytes, PARP overactivation was demonstrated to cause ΔΨm collapse, AIF release and nuclear translocation, and cell death, which all could be prevented by normalizing cytosolic NAD^+^ levels [153,154]. PARP activation is seemingly not essential for the observed ΔΨm collapse, AIF release, and cell death, since NAD^+^ depletion alone can induce all these effects. [154]. However, considering the copiousness of PARP1, normally its over-activation is most likely the cause of NAD^+^ depletion. This situation may likely occur in the penumbra after cerebral or myocardial ischemia, where, in an attempt to rescue the cells [155], reverse operation of the ATP synthase consumes ATP to maintain ΔΨm, and substrate-level phosphorylation provides the necessary ATPs by utilising non-glucose substrates [156,157]. However, the said mechanism has a limited capacity, providing a relatively narrow time-window to overcome the detrimental effects of PARP activation [155]. In addition to PARPs, cyclic ADP-ribose hydrolase (CD38) and sirtuins (SIRTs) use NAD^+^ as a substrate. However, sirtuins are unlikely to participate in the NAD^+^ depletion process due to their low affinity for the substrate. On the other hand, when PARP inhibition limits NAD^+^ consumption, the resulting SIRT1 activation enhances PGC-1α expression, thereby increasing mitochondrial biogenesis, metabolism, and antioxidant capacity [7]. The expression level of PGC-1α is seemingly crucial in resisting hypoxic conditions and oxidative stress situations, as it was revealed when comparing oxidative resistance of melanomas expressing PGC-1α in high and low levels [158].

### 5.3. Coupling Glycolysis and Oxidative Phosphorylation

The PI3K-Akt pathway can also regulate mitochondrial metabolism by directly phosphorylating hexokinase II, the mitochondrial form of this glycolytic enzyme. Phosphorylated hexokinase II associates with VDAC of the outer mitochondrial membrane coupling glycolysis to oxidative phosphorylation, which enables the cell to cope with low oxygen conditions better [159]. Proximity to the mitochondria enables glycolysis to utilise mitochondria-produced ATP for the priming reactions of the pathway (hexokinase and phosphofructokinase) and channel pyruvate directly to the mitochondria in support of ATP production. The result aids in maintaining ΔΨm and VDAC opening [160]. Akt also regulates the process by phosphorylating, and thereby inactivating GSK-3β, which phosphorylates hexokinase II, resulting in disruption of its association with VDAC [161]. PAR fragments also can uncouple glycolysis and oxidative phosphorylation by dissociating hexokinase from the outer mitochondrial membrane [162]. Disruption of this coupling decreases the efficacy of glycolysis, and changes the potential of the outer membrane, leading to cytochrome C release and apoptosis induction, as well as VDAC pore closing, which causes mitochondrial swelling and -rupture [163]. Additionally, excess PARP activation interferes with mitochondrial quality control processes such as mitophagy and unfolded protein response [164].

## 6. Therapeutic Implications

### 6.1. Sepsis and Septic shock

Sepsis is a dysregulated host response to severe infection resulting in multi-organ failure and a mortality of 20 to 50%, despite advances in intensive care [165]. In sepsis, due to elevated ROS production and reduced antioxidant capacity, there is a substantial oxidative stress. Within the cells, the reduced/oxidised glutathione ratio drops [166]. Elevated ROS and pro-inflammatory mediators directly damage mitochondrial DNA, lipids, and proteins, and interfere with the machinery of oxidative phosphorylation [167]. Large amounts of NO**˙** are produced by the inducible NOS that inhibits complex IV [166]. The elevated ROS and RNS induces DNA strand break-mediated PARP activation causing reduction in the cellular NAD^+^ level, oxidative phosphorylation abnormalities, and mPT induction [168]. All these processes lead to the release of various small molecules, such as ATP and succinate, and macromolecules, or their fragments, such as mitochondrial DNA, RNA, transcription factor, cytochrome C, cardiolipin and N-formyl peptides, collectively referred to as mitochondria-derived damage-associated molecular patterns (mtDAMPs) [169]. The mtDAMPs are recognised by their associated pattern recognition receptors that induce various inflammatory processes, overactivation of which is responsible for the development of sepsis and septic shock [170]. Due to mitochondrion’s prokaryotic origin, mtDAMPs are similar to molecular patterns of most pathogens, and are likely to trigger excessive inflammatory reactions, which can aggravate sepsis to multi-organ dysfunction syndrome and death [169]. Activation of the inflammatory processes by mtDAMP can further damage mitochondria, forming a vicious circle that is a potential target regarding therapy. Inflammasome and mtDAMP receptor antagonists are suggested for this purpose [170], although repurposing of FDA approved PARP inhibitors for sepsis therapy may be justified [4]. Results on the various experimental sepsis and endotoxemia models in animals are certainly promising, since PARP inhibitors in these models improved survival and diminished multi-organ failure [171].

### 6.2. Atherosclerosis and Myocardial Infarction

In atherosclerosis, the chronic inflammation, cell damage, and arterial vascular endothelial dysfunction initiated by a physical or chemical injury is exaggerated by vascular cell adhesion molecules-induced adherence of neutrophils, lymphocytes, and monocytes to the endothelial cells [172]. The resulting elevated permeability of the vascular endothelium to endothelial cells and smooth muscle cells is caused by endothelial glycocalyx disruption and inhibition of voltage-sensitive K^+^, and Ca^2+^-activated K^+^ channels. Gradual accumulation of lipid and smooth muscle cells in the vessel intima leads to formation of atherosclerotic plaques, which produce pro-inflammatory factors inducible NOS, and ROS, thereby aggravating the disease [173]. ONOO− formed from NO**˙** and O_2_^−^**˙** enters the nucleus of endothelial cells, and induces DNA strand break-mediated PARP1 activation [174]. In addition to contributing to the vascular inflammation and endothelial dysfunction, PARP1 can facilitate atherosclerotic plaque disruption by activating matrix metalloproteinases (MMPs) [174]. Moderate PARP activation leads to AIF or caspase-mediated apoptosis while overactivation causes atherosclerotic plaque disruption and necrotic cell death [172]. PARP inhibitors were demonstrated to ameliorate all these effects in various experimental systems [172,173,174].

Acute myocardial infarction is followed by reperfusion, due to either coronary thrombolytic therapy or surgical intervention [175]. However, the return of oxygen to a respiratory chain that has suffered hypoxic damages during the preceding ischemia results in a substantial perfusion injury characterised by massive ROS and RNS production, upregulated expression of inflammatory cytokines, and PARP activation [176]. Additionally, the oxidative and nitrosative stress leads to endothelial dysfunction, transendothelial migration of neutrophils, JNK and p38 MAPK activation, DNA fragmentation, and myocardial injury [174]. MMP-2 activated by ONOO− cleaves troponin I, which results in impaired myocardial contractility [177]. PARP1 activation is sustained long after the coronary reperfusion in the necrotic zone and the penumbra, resulting in impeded systolic and diastolic function of the left ventricle, Ca^2+^ overload, and eventual cardiomyocyte death [178]. PARP inhibition achieved either by pharmacological means or by genetic manipulation significantly diminished the said detrimental effects in various experimental models [133,173,174,176]. Hearts of PARP1-deficient mice were reported to resist ischemic myocardial depression as well as reperfusion injury [179]. In Langendorff-perfused isolated rat hearts, PARP inhibitors were demonstrated to improve ATP and creatine phosphate recovery, attenuate ischemia–reperfusion-induced lipid peroxidation, protein oxidation, DNA strand breaks, and inactivation of respiratory complexes [133]. In a porcine myocardial infarction model, the PARP inhibitor PJ34 was reported to reduce infarct size by 25.7% and forestall myocardial PAR formation in the area [180]. In a rat model of arterial restenosis following angioplasty, PJ34 reduces leukocyte infiltration, neointima formation, and carotid artery restenosis, and improves functional recovery [181]. Finally, in a human clinical study of 40 patients of ST segment elevation myocardial infarction, the PARP inhibitor INO-1001 diminishes serum PARP activity and reduced inflammation [182].

### 6.3. Stroke

Ischemic stroke is caused by occlusion or collapse of one or more arteries of the central nervous system (CNS), resulting in critically reduced blood flow in a region. Although energy demand of the CNS is high, neuron and glia cells can survive without oxygen up to several hours on glycolysis, despite its meagre energy yield of 2 ATP/glucose molecule—a possible scenario in the ischemic penumbra. However, they succumb within minutes once both oxygen and glucose are deprived—the situation in the core of the ischemic area [183]. Since the brain can exclusively utilise glucose regarding energy production, blocking glycolysis results in deprivation of fuel molecules (pyruvate) from the mitochondrial ATP-producing machinery (pyruvate dehydrogenase complex, citrate cycle, oxidative phosphorylation), causing complete energy failure, even in the presence of oxygen [155]. ATP shortage is aggravated by the temporary reverse operation of the ATP synthase, which consumes rather than produces ATP in a futile attempt to maintain ΔΨm until fuel for substrate-level phosphorylation runs out [49]. The resulting ATP depletion collapses the cell membrane potential, enabling the influx of extracellular Ca^2+^ via the voltage-gated Ca^2+^ channels and reversing the glutamate uptake transporters responsible for the majority of glutamate release during ischemia [184]. Reperfusion, which is restoration of the blood flow, still may kill the cells by excitotoxicity, oxidative stress, inflammatory processes, apoptosis, or parthanatos [183]. Excitotoxicity is caused by the sustained action of glutamate on primarily the N-methyl-D-aspartate type of glutamate receptors and results in ONOO− and ROS production-mediated DNA damage accumulation, which activate PARP1 [185]. However, PARP1-induced NAD^+^ depletion affects the mitochondrial NAD^+^ pool only, if opening of the high conductance PTP allows NAD^+^ traffic between the matrix and the cytosol [186]. Even the transient opening of the PTP that does not harm the inner and outer membranes permits NAD^+^ release from the mitochondria, indicating the pivotal importance of mPT in mediating ischemia–reperfusion-induced impairment of mitochondrial energy production [187]. As an experimental confirmation, focal ischemia decreases infarct volume dramatically in animals deficient in cyclophilin D [188], the matrix peptidyl-propyl cis-trans isomerase, which desensitise Ca^2+^-induced opening of the PTP [189]. To demonstrate their potential in human stroke therapy [4], PARP inhibitors were reported to reduce ROS and RNS production, decrease oxidative DNA damage, protein oxidation, and lipid peroxidation, diminish infarct volume, and improve mitochondrial functions in various animal models regarding hypoglycemia as well as focal and global ischemia [155,186].

### 6.4. Neuroegenarative Diseases

Alzheimer’s disease is a type of amyloidosis of the CNS, leading to progressive cognitive decline. Senile plaques of β-amyloid peptide and neurofibrillary tangles formed by hyperphosphorylated tau are the pathological hallmarks of the disease [190]. Pathogenesis of the disease involves oxidative mtDNA damage, leading to accumulation of mutations, decreased mtDNA transcription, and mtDNA copy number, eventually resulting in mitochondrial dysfunction [190]. The ensuing ROS production activates the Nrf-2 antioxidant defence pathway while the β-amyloid peptide causes mitochondrial fragmentation via DRP1 [191,192]. Inhibition of DRP1 prevented mitochondrial fragmentation, mitochondrial dysfunction, decreased β-secretase 1 expression and β-amyloid deposition, and improved cognitive function of transgenic mice bearing human familial Alzheimer’s disease genes [193,194].

Parkinson’s disease is characterised by motor dysfunction and dementia. The pathology features the loss of dopaminergic neurons and accumulation of α-synuclein deposits (Lewy bodies) in the substantia nigra [195]. Interaction of α-synuclein with respiratory complex I as well as outer mitochondrial membrane proteins TOM20 and VDAC, contributes to mitochondrial dysfunction, oxidative stress, and the resulting neuropathological changes [195]. The PTEN-induced putative kinase (PINK)1 and Parkin, an E3 ubiquitin ligase, participate in mitochondrial quality control via mitophagy. Their mutation leads to ubiquitinylation, then proteosomal degradation of Mfn, prevention of mitochondrial fusion, and premature mitophagy [196]. The protein deglycase DJ-1 translocates to the mitochondria upon oxidative stress-induced acidification of its Cys^106^, where it strengthens the antioxidant defence system by stabilising and preventing the degradation of Nrf-2 [197]. Activating mutation of the leucine-rich repeat kinase 2 increases DRP1 Ser^616^ phosphorylation, thereby activating mitochondrial fission and mitophagy [198]. High-temperature requirement protein A2 participates in the mitochondrial quality control by degrading denatured proteins inside the mitochondria. This intermembrane protease is released upon apoptotic stimuli, binds inhibitor of apoptosis proteins, then cleaves caspases and other proteins leading to apoptotic death [191].

Huntington’s disease, characterised by chorea, dystonia, incoordination, and cognitive decline, is caused by CAG trinucleotide repeat expansion in the huntingtin gene, manifesting in polyglutamine repeats in the huntingtin protein [199]. Mitochondrial respiration and ATP synthesis are limited by impeded operation of respiratory complexes II and III [200]. Mutant huntingtin reduces PGC-1α expression via binding to CREB and transcription factor IID subunit 4-activating transcription factors [201]. Additionally, increased Ca^2+^ influx activates transglutaminase-2 that reduces PGC-1α expression by stabilising chromosomal structure, resulting in mtDNA lesions and depletion. Mutant huntingtin activates DRP1 and FIS-1 while downregulates Mfn1/2, thereby facilitating mitochondrial fragmentation and mitophagy [192]. Also, it can promote apoptosis by interacting with p53 or CBP transcription factors and PUMA or Bax activation, respectively [202]. Currently, therapy is restricted to ameliorate symptoms and there are no disease-modifying treatments yet available. However, PARP-1 inhibition was reported to protect striatal interneurons in the R6/2 mouse model of Huntington’s disease [203,204].

## 7. Concluding Remarks

The intended extension of PARP inhibitors in cancer therapy aims at the non-homologous recombination DNA repair systems, signalling pathways, angiogenesis, or immune checkpoint mechanisms, all utilising the function of PARP in DNA repair. However, PARP activation affects multiple cellular processes in addition that may form the basis of repurposing PARP inhibitors for non-oncological indications [4]. In many diseases associated with oxidative stress, PARP activation has nuclear effects such as the NAD^+^ and ATP depletion-mediated necrosis and various extra-nuclear effects on cytosolic and mitochondrial processes. Although the pathomechanism of oxidative stress-associated diseases is multiform, some mitochondrial processes such as ROS production and mPT seemingly have a significant role in most of these diseases. Unfortunately, the said processes are initiated via multiple mechanisms in addition to PARP activation, therefore, inhibition of the enzyme does not necessarily prevent them. The combination of PARP inhibition with an antioxidant or mPT inhibiting agent may represent a solution, and cyclophilin D-deficient animals could serve as a useful experimental system in this respect. A substantial amount of preclinical data is available in reference to the beneficial effects of PARP inhibitors in cell culture, isolated organ or animal models of various oxidative stress-associated human diseases. However, they often fall outside the interest of investigators in the field of cancer research. Additionally, the translational power of such data is questionable. The models fail to recapture timescale and complexity of the disease, therefore only clinical studies can establish the realistic potentiality of PARP inhibitors in a given disease. Although the FDA approval of PARP inhibitors for human cancer therapy created the opportunity, there is an understandable reluctance to administer to patients a substance that regulates such a fundamental cellular process as DNA repair. However, a better understanding of the disease’s pathomechanisms, and accumulating data regarding combination treatments aiming at a coordinated, or ideally, synergistic effect will certainly aid in introducing PARP inhibitors to oxidative stress-associated diseases. Presenting recent data and models of PARP inhibitor-induced processes protecting the mitochondria against the detrimental effects of oxidative stress in this review may hopefully stimulate further preclinical and clinical research in this field.

## Figures and Tables

**Figure 1 ijms-21-02767-f001:**
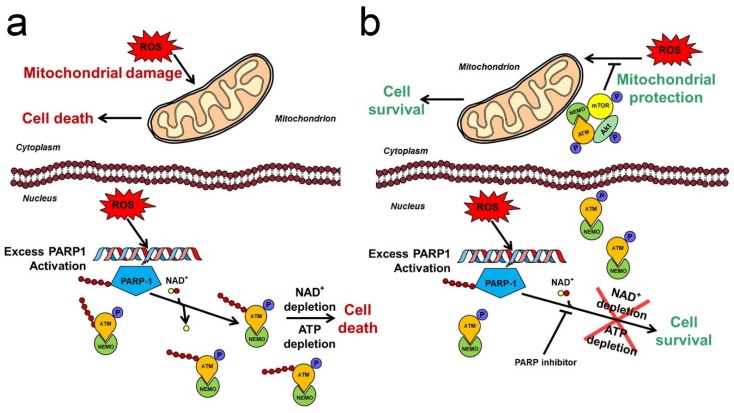
Interplay of poly(ADP-ribose) polymerase (PARP) with Akt-mediated mitochondrial protection in oxidative stress. (**a**) Effects of oxidative stress in the absence of PARP inhibition. ROS-induced DNA strand breaks induce excess PARP activation leading to NAD^+^ then ATP depletion. PARylated ATM–NEMO are retained in the nucleus. ROS-induced mitochondrial dysfunction contributes to cell death caused by energy failure. (**b**) Effects of oxidative stress in the presence of PARP inhibition. The PARP inhibitor blocks excess PARP activation and NAD^+^ consumption. Activated ATM–NEMO complex translocates to the cytoplasm and attaches to the outer mitochondrial membrane. Akt and mTOR are complexed with ATM–NEMO to form an ATM–NEMO–Akt–mTOR signallosome. Akt becomes activated and protect the mitochondria against the ROS-induced damages. Pointed arrows denote activation while arrows with flat end represent inhibition. P indicates phosphorylation. Red and yellow spots designate ADP-ribose and nicotinamide, respectively.

**Figure 2 ijms-21-02767-f002:**
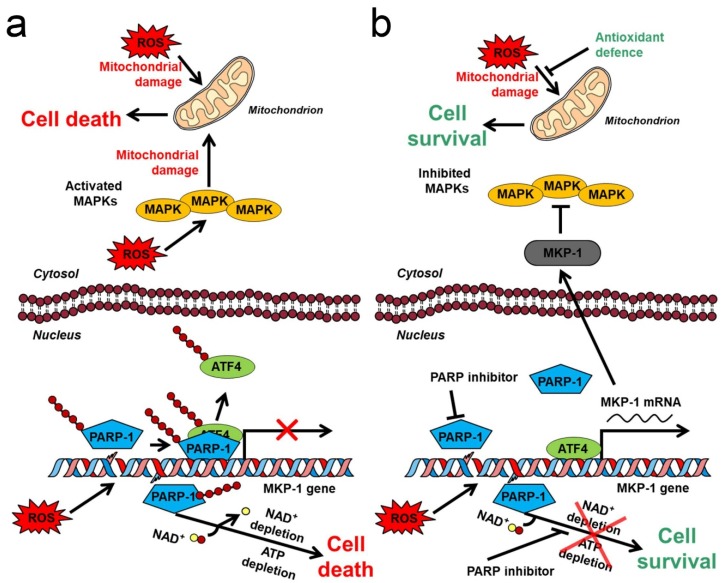
Interplay of PARP with MAPK-mediated mitochondrial damage in oxidative stress. (**a**) Effects of oxidative stress in the absence of PARP inhibition. ROS-induced DNA strand breaks induce excess PARP activation leading to NAD^+^, then ATP depletion. PARP PARylates both itself and ATF4, and replaces PARylated ATM4 at the promoter of MKP-1 coding DNA region. ROS activates JNK and p38 MAPK, which exaggerate ROS-induced mitochondrial dysfunction, and contributes to cell death caused by energy failure. (**b**) Effects of oxidative stress in the presence of PARP inhibition. The PARP inhibitor blocks excess PARP activation and NAD^+^ consumption. ATF4 is not PARylated, therefore is able to bind to the promoter of MKP-1 coding DNA region. MKP-1 mRNA is transcribed, MKP-1 protein is translated and inactivates JNK and p38 MAPK. The antioxidant defences compensate for the ROS-induced mitochondrial damages. Pointed arrows denote activation while arrows with flat end represent inhibition. P indicates phosphorylation. Red and yellow spots designate ADP-ribose and nicotinamide, respectively.

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
