# Peer review of "Mitochondrial Protection by PARP Inhibition"

_ijms, 2020, doi:10.3390/ijms21082767_

Round 1
Reviewer 1 Report
Review: Mitochondrial Protection by PARP Inhibition:
Therapeutic Considerations
The article is interesting; it is armed with the perspective of summarizing mitochondrial basics activity, focusing on its structure, function and also regarding oxidative stress.
As for PARP and its inhibitors it is explained, although I do not consider it to comply with the proposed title, I think it is ambitious especially when it comes to "Therapeutic Considerations"
The figures help quite a lot to interpret changes proposed. However, the conclusions do not reflect all the aspects seen in this review. They should be properly reviewed and completed.
Author Response
We thank the review his/her thorough work. We think that his/her contribution has improved the manuscript considerably.
The reviewer rightly observed that the topic was overambitious, especially regarding therapeutic considerations. Previously, a very comprehensive review summarized the preclinical data supporting the repurposing of PARP inhibitors in non-oncological diseases. In that review (Berger et al, 2018; reference 4), therapeutic risks as well as demands were evaluated by leading authorities in the field. Therefore, instead of the preclinical data and therapeutic implications, we intended to focus on the mechanisms of PARP-activation induced mitochondrial damages. Accordingly, we removed “Therapeutic Considerations” from the title, and modified the Abstract, Introduction and Conclusion sections of the manuscript to clarify this intention.
We fully agree with the reviewer that conclusions do not reflect all the aspects seen in this review. Actually, we did not feel confident to draw definite conclusions, that is why we changed title of the section to “Concluding remarks”. In an attempt to comply with the reviewer’s request, we added a paragraph about the role of reactive oxygen species (ROS) production and mitochondrial permeability transition (mPT) in mitochondrial malfunction, and the potentiality of cyclophilin D deficient mice as an experimental system for collecting preclinical data about combination of PARP inhibitors with antioxidants or mPT inhibitors.
Reviewer 2 Report
The intensive review of mitochondrial injury and protection related with PARP and its inhibitor is expected to be very helpful to researchers.
Although detailed discussion is deep enough, generally the introductions of the major sections seem too long to reach focused discussion about PARP and PARP inibition.
Terms like A-induced, B-mediated, C-associated, D-generated , E-related .... are consistently used without "-". Please check these styles are correct.
There are some grammar errors
Author Response
We thank the review his/her thorough work. We think that his/her contribution has improved the manuscript considerably.
The reviewer rightly observed that the introductions of the major sections are too long to reach focused discussion about PARP and PARP inhibition. In fact, the theme of the review has proved to be too ambitious, especially regarding therapeutic considerations. Previously, a very comprehensive review summarized the preclinical data supporting the repurposing of PARP inhibitors in non-oncological diseases. In that review (Berger et al, 2018; reference 4), therapeutic risks as well as demands were evaluated by leading authorities in the field. Therefore, instead of the preclinical data and therapeutic implications, we intended to focus on the mechanisms of PARP-activation induced mitochondrial damages. Accordingly, we removed “Therapeutic Considerations” from the title, and modified the Abstract, Introduction and Conclusion sections of the manuscript to clarify this intention.
We corrected all implicated expressions.
The tight deadline did not allow us to contact a professional proof-reading agency, therefore we asked for the services of the Medical School’s in-house proof-reader, a native speaker USA citizen, who is unfortunately not a professional in the field. Therefore, a number of the corrections suggested by him were not accepted by us, since they have resulted from misunderstanding science expressions, and differences between UK and USA English. However, we do hope that our collective effort has eliminated grammatical errors from the revised manuscript as requested by the reviewer.